# ZnO Nanorods Create a Hypoxic State with Induction of HIF-1 and EPAS1, Autophagy, and Mitophagy in Cancer and Non-Cancer Cells

**DOI:** 10.3390/ijms24086971

**Published:** 2023-04-09

**Authors:** Michele Aventaggiato, Adele Preziosi, Hossein Cheraghi Bidsorkhi, Emily Schifano, Simone Vespa, Stefania Mardente, Alessandra Zicari, Daniela Uccelletti, Patrizia Mancini, Lavinia Vittoria Lotti, Maria Sabrina Sarto, Marco Tafani

**Affiliations:** 1Department of Experimental Medicine, Sapienza University, Viale Regina Elena 324, 00161 Rome, Italy; michele.aventaggiato@uniroma1.it (M.A.); stefania.mardente@uniroma1.it (S.M.); alessandra.zicari@uniroma1.it (A.Z.); patrizia.mancini@uniroma1.it (P.M.); laviniavittoria.lotti@uniroma1.it (L.V.L.); 2Department of Biology and Biotechnology “Charles Darwin”, Sapienza University, P.le A. Moro,5, 00185 Rome, Italy; adele.preziosi@uniroma1.it (A.P.); emily.schifano@uniroma1.it (E.S.); daniela.uccelletti@uniroma1.it (D.U.); 3Department of Aerospace, Electrical and Energy Engineering, Sapienza University, Via Eudossiana 18, 00184 Rome, Italy; hossein.cheraghibidsorkhi@uniroma1.it (H.C.B.); mariasabrina.sarto@uniroma1.it (M.S.S.); 4Research Center for Nanotechnology Applied to Engineering, Sapienza University, Via Eudossiana 18, 00184 Rome, Italy; 5Center for Advanced Studies and Technology, University “G. D’Annunzio” of Chieti-Pescara, Via Luigi Polacchi 11, 66100 Chieti, Italy; simone.vespa@unich.it

**Keywords:** ZnO nanorods, hypoxia, autophagy, mitophagy, nanomaterials, cancer

## Abstract

Nanomaterials are gaining increasing attention as innovative materials in medicine. Among nanomaterials, zinc oxide (ZnO) nanostructures are particularly appealing because of their opto-electrical, antimicrobial, and photochemical properties. Although ZnO is recognized as a safe material and the Zn ion (Zn^2+^) concentration is strictly regulated at a cellular and systemic level, different studies have demonstrated cellular toxicity of ZnO nanoparticles (ZnO-NPs) and ZnO nanorods (ZnO-NRs). Recently, ZnO-NP toxicity has been shown to depend on the intracellular accumulation of ROS, activation of autophagy and mitophagy, as well as stabilization and accumulation of hypoxia-inducible factor-1α (HIF-1α) protein. However, if the same pathway is also activated by ZnO-NRs and how non-cancer cells respond to ZnO-NR treatment, are still unknown. To answer to these questions, we treated epithelial HaCaT and breast cancer MCF-7 cells with different ZnO-NR concentrations. Our results showed that ZnO-NR treatments increased cell death through ROS accumulation, HIF-1α and endothelial PAS domain protein 1 (EPAS1) activation, and induction of autophagy and mitophagy in both cell lines. These results, while on one side, confirmed that ZnO-NRs can be used to reduce cancer growth, on the other side, raised some concerns on the activation of a hypoxic response in normal cells that, in the long run, could induce cellular transformation.

## 1. Introduction

Nanomedicine is a new field of study that deals with innovative materials in a variety of medical applications and nanomaterials are the most encouraging alternatives to meet the requirements of nanomedicine, with their typical sizes being on the nanometer scale. Due to the number of atoms on the exposed surface, nanomaterials have a large impact on biological systems [1].

At present, among various nanomaterials, zinc oxide nanostructures have attracted more and more attention over the past decades because of their opto-electrical, antimicrobial (Gram-positive, Gram-negative, and fungi), and photochemical properties [2,3,4,5]. ZnO has been recognized as a safe compound by the US Food and Drug Administration (“Generally Recognized As Safe (GRAS)”) and is the second most abundant metal oxide in biosystems. ZnO is vital for various cellular mechanisms as it helps maintaining cellular homeostasis, and hence is biocompatible [6]. ZnO is a semiconductor with a wide direct-band gap of 3.3 eV. ZnO and ZnO-related materials are widely utilized in various fields such as use in photocatalysts, semiconductors, solar cells, and pharmaceuticals [7,8,9,10].

Among various ZnO nanomaterials, ZnO-NRs show a wurtzite crystal structure, which is formed by interpenetrating, closely packed, hexagonal sub-lattices consisting of one specific kind of atom. The wurtzite structure of ZnO results in higher chemical and mechanical stability [11].

The shape and size of ZnO nanostructures impact their cytotoxicity with unexpected consequences for human health [12]. It has been shown that long ZnO-NRs exhibit lower toxicity than short ZnO-NRs and ZnO-NPs [13]. In another study, ZnO-NRs of different sizes (300 nm, 1 μm, and 5 μm), prepared via a solvothermal method, induced both dose- and size-dependent cellular responses correlating with a production of reactive oxygen species (ROS) [14]. Macrophage adhesive response and viability were also evaluated when cells were treated with ZnO-NRs in comparison with a sputtered ZnO layer on a glass substrate as a control [15]. Neonatal rat cardiomyocytes were challenged with ZnO-NRs either as a substrate or patch resulting in the inhibition of cell adhesion and spreading when used as a substrate and killing the cells underneath when used as a patch [16]. Several works report an antitumor effect of ZnO nanoparticles, and recently, mitochondria-targeted ZnO-NRs were used for an efficient breast cancer treatment [17].

The antitumoral effect of ZnO nanoparticles on osteosarcoma has been ascribed to the activation of autophagy and mitophagy due to the stabilization and accumulation of HIF-1α protein by both ROS and Zn^2+^ [18]. Moreover, such an effect would lead to the degradation of β-catenin, preventing osteosarcoma metastasis [19]. The same effects were not studied in non-cancerous cell lines but only in tumor cells or in xenografts leaving open the question if the same molecular pathways are also activated by ZnO-NRs. For this reason, our work aims to investigate the following aspects: (i) compare the biological effects of ZnO-NRs in non-tumor (HaCaT, epithelial) and tumor (MCF-7, breast cancer) cell lines, and (ii) determine if a similar or different molecular mechanism is elicited by ZnO-NRs in these two cell lines.

## 2. Results

### 2.1. ZnO-NRs Reduce Cell Vitality and Colony Formation

ZnO-NR were synthesized as described in the Materials and Methods section. Field Emission Scanning Electron Microscopy (FE-SEM) imaging was performed on the produced materials using a Zeiss Auriga SEM, operating at an accelerating voltage of 5 keV. From the FE-SEM analysis, it was found that the ZnO-NRs have diameters between 40 and 50 nm and lengths that can approach 800 nm after sonication (Figure 1).

In order to determine the toxicity of ZnO-NRs, cancerous (MCF-7) and non-cancerous (HaCaT) cells were treated with different concentrations of ZnO-NRs up to 48 h.

Figure 2A,B shows that the ZnO-NRs reduced the cell vitality of both HaCaT and MCF7 cells in a dose-dependent manner after 24 h. These effects were maintained at both ZnO-NR concentrations in both cell lines for 48 h (Figure 2A,B). Similarly, colony formation was reduced by treatment of both cell lines with ZnO-NRs (Figure 2C,D). However, colony inhibition was higher in HaCaT than in MCF7 cells (Figure 2C,D). Although one proposed mechanism for ZnO-NRs’ toxic effect on cell lines is through a significant increase in ROS [20], in our experimental setting we only found a slight increase in ROS levels after ZnO-NR treatment in both cell lines as shown in Figure 2E. Finally, we also determined the concentration of intracellular Zn^2+^ after ZnO-NR treatment. As shown in Figure 2F, intracellular Zn slightly increased in HaCaT cells treated with 20 µg/mL ZnO-NRs for 48 h and with 50 µg/mL for 24 and 48 h. An intracellular Zn increase was also observed in MCF-7 cells treated for 48 h with 50 µg/mL ZnO-NRs.

### 2.2. ZnO-NRs Create a Hypoxic State with HIF-1 and EPAS1 Activation

Recently, ZnO-NPs have been shown to activate HIF-1α by both ROS and Zn intracellular accumulation [18]. To determine if ZnO-NRs also activate the same response and to visualize the distribution of ZnO-NRs in our cell lines, we performed TEM analyses. The results in Figure 3 show that ZnO-NRs accumulate mostly on top of the plasma membrane in both HaCaT and MCF-7 cells.

Along with the plasma membrane localization, we also observed intracellular accumulation of ZnO-NRs in vesicles just below the membrane as well as in the cytoplasm as indicated by the black arrows (Figure 3B,C,E,F). To test if such extracellular deposition and intracellular accumulation of ZnO-NRs create a hypoxic state in the cells, we measured expression as well as intracellular localization of HIF-1α and EPAS1 (HIF-2). The results showed that both 20 and 50 µg/mL ZnO-NRs increased HIF-1α protein accumulation and its nuclear translocation in HaCaT cells after 24 and 48 h (Figure 4A–E).

Moreover, we determined the expression of HIF-1α target genes *HKII* and *CAIX*. Figure 4C,D,F,G shows the increase in HKII expression after 24 h treatment of HaCaT cells with 50 µg/mL ZnO-NRs. Conversely, we observed a decrease in CAIX expression after 24 and 48 h with 20 and 50 µg/mL ZnO-NRs (Figure 4G). It must be noted that the same trend of increased HKII after 24 h and decreased CAIX after 48 h was obtained with the CoCl_2_ treatment used as a positive hypoxic control. In the case of MCF-7 cells, we observed a similar increase in HIF-1α expression. However, HIF-1α localization was mostly perinuclear and was not accompanied by nuclear translocation (Figure 5A–E).

Moreover, the analysis of HIF-1α target genes revealed an increase in HKII protein levels after 24 h and CAIX after 48 h of ZnO-NR treatment (Figure 5C,D,F,G). Since the hypoxia response is also controlled by EPAS1 that is expressed by MCF-7 cells [21], we studied EPAS1 expression and its intracellular localization after ZnO-NR treatment. The results showed that HaCaT cells have a low perinuclear expression of EPAS1 whose increase is concentration-dependent at 24 h, while at 48 h EPAS1 expression tended to decrease at lower concentrations of ZnO-NRs (10 and 20 µg/mL) (Figure 6A,B) and showed no variation compared with control cells at 50 µg/mL ZnO-NRs (Figure 6C–E).

MCF-7 cells have a high basal EPAS1 expression level that is further increased by ZnO-NRs after 24 h and 48 h (Figure 7).

### 2.3. Autophagy and Mitophagy Regulation by ZnO-NRs

Hypoxia has been shown to activate autophagy and mitophagy in different cellular systems [22,23]. Moreover, activation of autophagy and mitophagy has been observed after ZnO-NP treatment in osteosarcoma [19].

Figure 8A shows that in HaCaT epithelial cells, ZnO-NR treatment increased mitophagy marker Bcl2 interacting protein 3 (BNIP3) and autophagy marker microtubule-associated protein 1A/1B light chain 3 (LC3II) after 24 h. TEM analysis after 24 h of ZnO-NR treatment, confirmed the presence of mitophagic and autophagic vacuoles indicated by the arrows in Figure 8B,C, respectively. Extending the ZnO-NR treatment to 48 h maintained the increase in BNIP3 levels only with the 50 µg/mL dose while decreasing LC3II levels (Figure 8D), an effect probably due to the toxicity of this ZnO-NR concentration. Expression of the apoptotic marker PARP showed that ZnO-NR treatment did not affect PARP cleavage after 24 h (Figure 8E) while it decreased it after 48 h at 50 µg/mL ZnO-NRs (Figure 8F). In the case of breast cancer MCF-7cells, we documented a marked increase in the expression of the mitophagy marker BNIP3 after treating the cells with 50 µg/mL ZnO-NRs for 24 h (Figure 9A).

No increase in LC3II expression was observed at the same time points and ZnO-NR concentrations (Figure 9A). TEM analysis after 24 h of ZnO-NR treatment revealed the presence of mitochondria being surrounded by mitophagic membranes as well as of mitophagic vacuoles (Figure 9B). We also observed new forming autophagic structures that were not completely closed (Figure 9C). Extending the treatment to 48 h resulted in an increase in BNIP3 with 20 µg/mL and a decrease with 50 µg/mL (Figure 9D). At the same time, we documented an increase in LC3II with 10, 20, and 50 µg/mL ZnO-NRs (Figure 9D). Finally, no increase in the cleavage of the apoptotic marker PARP was measured after ZnO-NR treatment (Figure 9E,F).

## 3. Discussion

In the present study, breast cancer (MCF-7) and non-tumor epithelial (HaCaT) cell lines were treated with different concentration of ZnO-NRs to evaluate toxicity and the underlying molecular mechanisms. Apart from minor differences in the extent of the cellular response, we generally did not observe major differences in the cellular and molecular response of these two cell lines to the ZnO-NR treatment. In fact, our results evidenced a reduction of cell viability and clonogenicity in both cell lines (Figure 2A–D). Indeed, while few studies investigated the cellular effects of ZnO-NRs, ZnO-NP toxicity has been reported for kidney, epithelial, and microglial cell lines [20,24,25,26]. On the other hand, ZnO-NPs also increase tumor cell death by different mechanisms. In particular, ZnO-NPs reduced vitality in neuroblastoma and colon carcinoma cells by activating apoptosis [27,28] and in osteosarcoma by activating autophagy [19]. Interestingly, the underlying molecular mechanisms elicited by ZnO-NPs seem to be activated by the accumulation of ROS and Zn^2+^ inside the cells [18]. Such ROS and Zn^2+^ increases have been linked to the activation of a hypoxic response with HIF-1α stabilization and accumulation [18,19]. In line with our previous results [24], we observed a slight increase in intracellular ROS and Zn^2+^ levels (Figure 2E,F). The intracellular Zn^2+^ increase, although statistically significant, was minimal and was probably due to the low amount of Zn^2+^ released by ZnO-NRs [29] and for the ample Zn-buffering capacity of the cells [6,30]. Therefore, we think that in the case of ZnO-NRs, free Zn^2+^ has almost no role in the cellular and molecular effects that we observed. Interestingly, we found, by TEM, the deposition of ZnO-NRs on the plasma membrane of MCF-7 and HaCaT cells (Figure 3), an event that we speculate may increase the onset of a hypoxic state. In fact, HIF-1α protein accumulation and nuclear translocation were increased following ZnO-NR treatment (Figure 4 and Figure 5). In turn, the accumulated HIF-1α increased transcription of target genes such as *HKII* and *CAIX* (Figure 4 and Figure 5). Interestingly, our study, for the first time, also demonstrated an increase in EPAS1 (HIF-2) in the two cell lines after ZnO-NR treatment (Figure 6 and Figure 7). However, EPAS1 expression was higher in MCF-7 than in HaCaT cells. Moreover, there was a significant EPAS1 increase in both HaCaT and MCF-7 cells during the first 24 h of ZnO-NR treatment followed by a decrease below the control only in HaCaT cells (Figure 6 and Figure 7). This agrees with results reported by other researchers showing increased EPAS1 expression in MCF-7 cells [21]. In osteosarcoma cells, ZnO-NP-induced HIF-1α activation has been linked to autophagy and mitophagy induction and cell death [18]. In our system, we observed autophagy and mitophagy induction. However, we documented a more rapid induction of the mitophagy marker BNIP3 in the breast cancer cells compared to the keratinocytes (Figure 8 and Figure 9). The activation of autophagy and mitophagy was accompanied by inhibition of apoptosis as documented by the fact that PARP cleavage was not increased by ZnO-NR treatment (Figure 8 and Figure 9).

## 4. Materials and Methods

### 4.1. ZnO-NR Synthesis and Characterization

All chemicals and reagents used in this study, except otherwise stated, were purchased from Sigma-Aldrich—MERCK St. Louis, MO, USA.

The synthesis of ZnO-NRs was achieved via thermal degradation according to [31] with minor modifications. The synthesis required a uniform distribution of 0.5 g of zinc acetate dihydrate (383058) inside a tightly sealed steel container. The steel container was then heated in a muffle furnace at 300 °C in air for 12 h, at which point, porous sheets of ZnO-NRs were created. The resulting ZnO-NRs were then allowed to cool before being dispersed in DMEM just before use.

### 4.2. Cell Culture and Treatments

The human keratinocyte cell line HaCaT, obtained from Thermo Fisher Scientific (Waltham, MA, USA), and the human breast cancer cell line MCF-7, obtained from American Type Culture Collection (ATCC, Manassas, VA, USA), were cultured in Dulbecco’s Modified Eagle’s Medium (DMEM; Euroclone, Milan, Italy) supplemented with 10% heat inactivated Fetal Bovine Serum (F9665), 2 mM L-glutamine (G7513), 100 units/mL penicillin, and 0.1 mg/mL streptomycin (P0781). Adherent cells were detached using a Trypsin-EDTA solution (T4049).

The suspensions of ZnO-NRs were freshly prepared in DMEM followed by 30 min bath-sonication, and were subsequently diluted to the final concentration indicated in the figures.

Cobalt(II) chloride hexahydrate (C8661) was dissolved in distilled, sterile water and added to a final concentration of 200 µM for 24 h or 48 h before harvesting the cells.

### 4.3. Cell Viability Assay

Cell viability was determined using the MTT assay as previously described [32]. Briefly, 3 × 10^4^ HaCaT and MCF-7 cells were plated in 24-well culture plates and cultured overnight. Subsequently, the cells were incubated with 0, 10, 20 and 50 µg/mL ZnO-NRs in DMEM and cultured for 24 or 48 h. Tetrazolium salts (MTT: 3-(4,5-dimethylthiazol-2-yl)-2,5-diphenyltetrazolium bromide, 5 mg/mL suspended in PBS) were added to each well and incubated for 4h. The formazan crystals were extracted from the cells with a solubilizing solution (DMSO). An ELISA reader (Multiskan™ FC Microplate Photometer, Thermo Fisher Scientific) was used to measure the absorbance at a wavelength of 570 nm and reference length of 630 nm. The results were expressed as a percentage of the viability of untreated cells.

### 4.4. Clonogenic Assay

Treated or untreated cells (5 × 10^2^) were plated in 100 mm dishes to perform the clonogenic assay. After the colonies were grown (11 days for HaCaT and 10 days for MCF-7), the plates were washed twice with a phosphate-buffered saline solution (PBS; 79382) and fixed with a 4% formaldehyde (F8775) solution in PBS at room temperature (RT). After 20 min, the plates were washed twice in PBS and stained for 5 min with 0.5% crystal violet solution (C0775). Finally, the cells were washed with distilled water and air-dried. The colonies were counted the following day.

### 4.5. Zn^2+^ Accumulation

The accumulation of Zn^2+^ inside cells was evaluated using Zinquin ethyl ester (Z2251), a cell-permeable Zn ion fluorescent probe. This compound was dissolved in dimethyl sulfoxide (DMSO; D2438) in a stock solution of 2.4 mM according to manufacturer’s instructions. After treatments, cells were washed twice with Hanks’ solution and incubated with Zinquin ethyl ester at a final concentration of 24 μM for 30 min. After one more wash, the cells were detached using Trypsin-EDTA, centrifuged at 555× *g* for 10 min, and counted by trypan blue exclusion assay. Briefly, cells were stained with 0.4% trypan blue (T8154) and the cell suspension was applied to a hemocytometer and counted using a phase contrast microscopy (NIKON EclipseTE2000U, Nikon Netherlands, Amsterdam, The Netherlands). Approximately 5 × 10^4^ cells were placed in a 96-well plate and Zinquin fluorescence at 365/410-460 was measured using a Glomax^®^-Multi Detection System (Promega, Milan, Italy).

### 4.6. Measurement of Reactive Oxygen Species

The cells were seeded and treated as previously described. At the end of the treatments, the cells were washed twice with PBS and incubated with 2′,7′-dichlorofluorescin diacetate (D6883) dissolved in DMSO at a final concentration of 10 μM for 30 min. After two washes in PBS, the cells were harvested, counted, and 5 × 10^4^ cells were placed in a 96-well plate. DCFH-DA fluorescence at 490/510-570 was measured using a Glomax^®^-Multi Detection System (Promega).

### 4.7. Western Blot Analysis

To obtain whole cell lysates, untreated and treated cells were harvested and centrifuged at 555× *g* for 10 min at 4 °C and the pellets were resuspended in a lysis solution containing 50 mM Tris-Cl (Tris-Cl; 93352), 250 mM sodium chloride (NaCl; S7653), 0.1% Triton^®^ X-100, 5 mM ethylenediaminetetraacetic acid (EDTA; E6758), 0.1 mM dithiothreitol (DTT; D9163), 1 mM phenylmethylsulfonyl fluoride (PMSF; 93482), 1 mM sodium orthovanadate (Na_3_VO_4_; S6508), 10 mM sodium fluoride (NaF; 201154), and protease inhibitor cocktail (PI; P8340). The samples were agitated on a vortex every 10 min three times and after 30 min of incubation on ice, they were centrifuged at 14,000× *g* for 10 min at 4 °C and the supernatants were collected. Protein concentration was determined by Bradford assay (500-0205, Bio-Rad, Hercules, CA, USA). Clarified protein lysates (50 μg) were boiled for 5 min, electrophoresed onto denaturating SDS-PAGE gels and transferred onto 0.45 µm nitrocellulose membranes (162-0115, Bio-Rad). The blotting membranes were blocked with 5% non-fat dry milk (1706404, Bio-Rad) for 1 h at RT and then incubated with primary antibody overnight at 4 °C. The following day, the membranes were washed three times with 0.1% Tween^®^ 20 in PBS (PBST) for 30 min at RT and incubated with the appropriate secondary antibody for 1 h at RT. After another 3 washes in PBST, the detection of the relevant protein was assessed by enhanced chemiluminescence (Lite Ablot^®^ TURBO; EMP012001 EuroClone, Milan, Italy). Densitometric analysis was performed using Image J Software version 1.53t (National Institutes of Health, Bethesda, MD, USA) and β-actin protein was used as the loading control. The following primary antibodies were used: β-ACTIN (A5316), HIF-1α (Cell Signaling; 14179), HIF2/EPAS-1 (sc-13596, Santa Cruz Biotechnology, Dallas, TX, USA), HKII (sc-6521, Santa Cruz Biotechnology), CAIX (NB100-417, Novus Biologicals Centennial, Centennial, CO, USA), BNIP3 (sc-56167, Santa Cruz Biotechnology), LC-3 (NB600-1384, Novus Biologicals), and PARP1 (sc-7150, Santa Cruz Biotechnology). The following secondary antibodies were used: Peroxidase-Conjugated AffiniPure Goat Anti-Rabbit IgG (H+L) (111-035-003, Jackson Immuno Research, Cambridgeshire, UK) and Peroxidase-Conjugated AffiniPure Goat Anti-Mouse IgG (H+L) (115-035-062, Jackson Immuno Research).

### 4.8. Immunofluorescence Microscopy Analysis

For immunofluorescence analyses, 3 × 10^4^ HaCaT and MCF-7 cells were seeded on glass coverslips and left to adhere. Then, the cells were exposed to 20 and 50 µg/mL ZnO-NRs in DMEM and to 200 µM CoCl_2_ for 24 and 48 h. Afterwards, the cells were fixed with 4% paraformaldehyde for 30 min, followed by treatment with 0.1 M glycine in PBS for 30 min and with 0.1% Triton X-100 in PBS for an additional 5 min, to allow permeabilization. HaCaT and MCF-7 cells were incubated with mouse anti-HIF-1α antibody (BD Bioscience, San Jose, CA, USA), 1:100 in PBS for 1 h, or with EPAS-1 ((190b): sc-13596, Santa Cruz Biotechnology), 1:50 for 1 h, followed by goat anti-mouse Alexa fluor 594 (Biotium, Fremont, CA, USA) 1:100 for 30 min. The nuclei were stained with 4′,6-diamidino-2-phenylindole (DAPI). The coverslips were finally mounted with Mowiol for observation. Immunofluorescence signals were analyzed with an Axio Observer Z1 inverted microscope equipped with an ApoTome.2 System (Carl Zeiss Inc., Oberkochen, Germany). Digital images were acquired with the AxioCam MRm high-resolution digital camera (Carl Zeiss Inc.) and processed with the AxioVision 4.8.2 software (Carl Zeiss Inc.).

### 4.9. TEM

The cells cultured as described above were fixed in 2% glutaraldehyde (G7651) in 0.1 M sodium cacodylate buffer pH 7.3 (C0250) for 24 h at 4 °C. After collection, the cells were washed three times in cacodylate buffer, post-fixed for 2 h in 1% OsO_4_ (75632), dehydrated in graded acetone solutions, and embedded in Epon-812 (Electron Microscopy Science, Hatfield, PA, USA). Ultrathin sections (60 nm) were cut using a Reichert ultramicrotome (Leica Microsystems, Wetzlar, Germany), counterstained with uranyl-acetate (EMS #22405; Electron Microscopy Science) and lead citrate, and finally examined using a Philips CM10 (TEM) and/or Fei-Philips Morgagni 268D transmission electron microscope (FEI, Eindhoven, The Netherlands).

### 4.10. Statistical Analysis

The data were expressed as the mean values ± standard deviations (SD) from at least three independent experiments. The statistical significance was determined by two-way ANOVA analysis coupled with a Bonferroni post-test (GraphPad Prism 11.0 software, GraphPad Software Inc., La Jolla, CA, USA) and defined as * *p* < 0.05, ** *p* < 0.01, and *** *p* < 0.001. Ns, not significant.

## 5. Conclusions

In conclusion, our results demonstrate that ZnO-NRs reduce cell vitality and clonogenicity in a similar way in HaCaT and MCF-7 cells. Moreover, a similar response was also present at the molecular level where ZnO-NRs activate a hypoxic response with HIF-1α and EPAS1 accumulation, increased HKII and CAIX, and mitophagy and autophagy activation in both cell lines. Our results suggest that HIF-1α and EPAS1 play a central role in the response of mammalian cells to ZnO-NRs, activating a series of pathways that should be the subject of future investigations. In fact, even if few studies, including ours, suggest that ZnO-NRs show anti-cancer effects through the activation of a hypoxic response, the fact that the same response is also elicited in non-tumor cells raises some concerns because it is known that a hypoxic state can be cytotoxic and genotoxic for normal cells and for tumor progression through the selection of more aggressive cells and clones or for the creation of hypoxic niches where cancer stem cells reside.

## Figures and Tables

**Figure 1 ijms-24-06971-f001:**
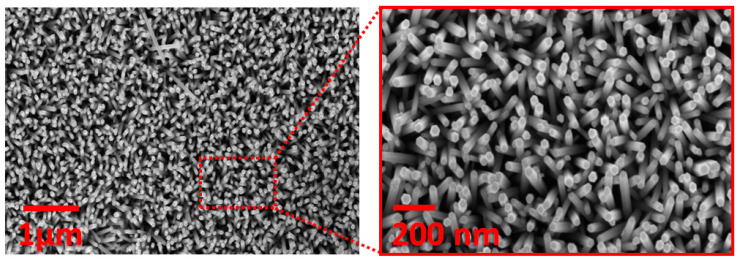
FE-SEM images of ZnO-NRs. FE-SEM images of hydrothermally grown ZnO-NRs, showing typical dimensions ranging between 40 and 50 nm in diameter, and lengths of ∼800 nm after sonication.

**Figure 2 ijms-24-06971-f002:**
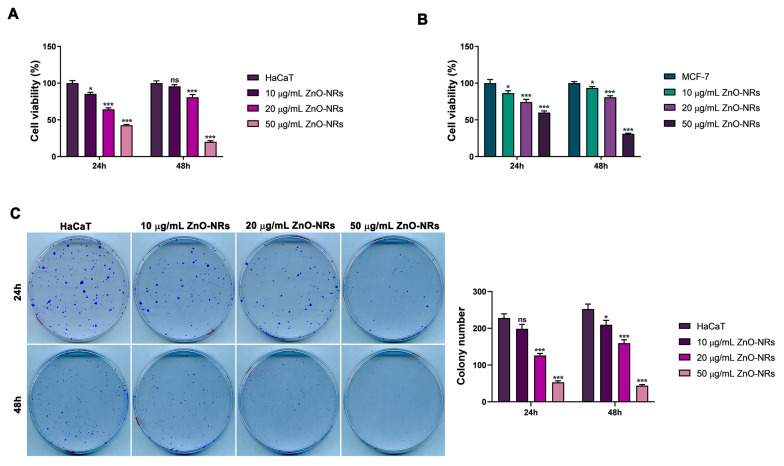
ZnO-NRs’ effects on HaCaT and MCF-7 cells. HaCaT and MCF-7 cells were either untreated or treated with the indicated ZnO-NR concentration for 24 and 48 h. The ncrease of ZnO-NR concentration caused a decrease (**A**,**B**) in cell viability and (**C**,**D**) clonogenicity ability. (**E**) Intracellular ROS measured in both cell lines after ZnO-NR treatment using 2′,7′-dichlorofluorescin diacetate (DCFH-DA). (**F**) Intracellular Zn^2+^ measured after ZnO-NR treatment using Zinquin. Experiments were repeated three times. * *p* < 0.05, ** *p* < 0.01, and *** *p* < 0.001. ns, not significant.

**Figure 3 ijms-24-06971-f003:**
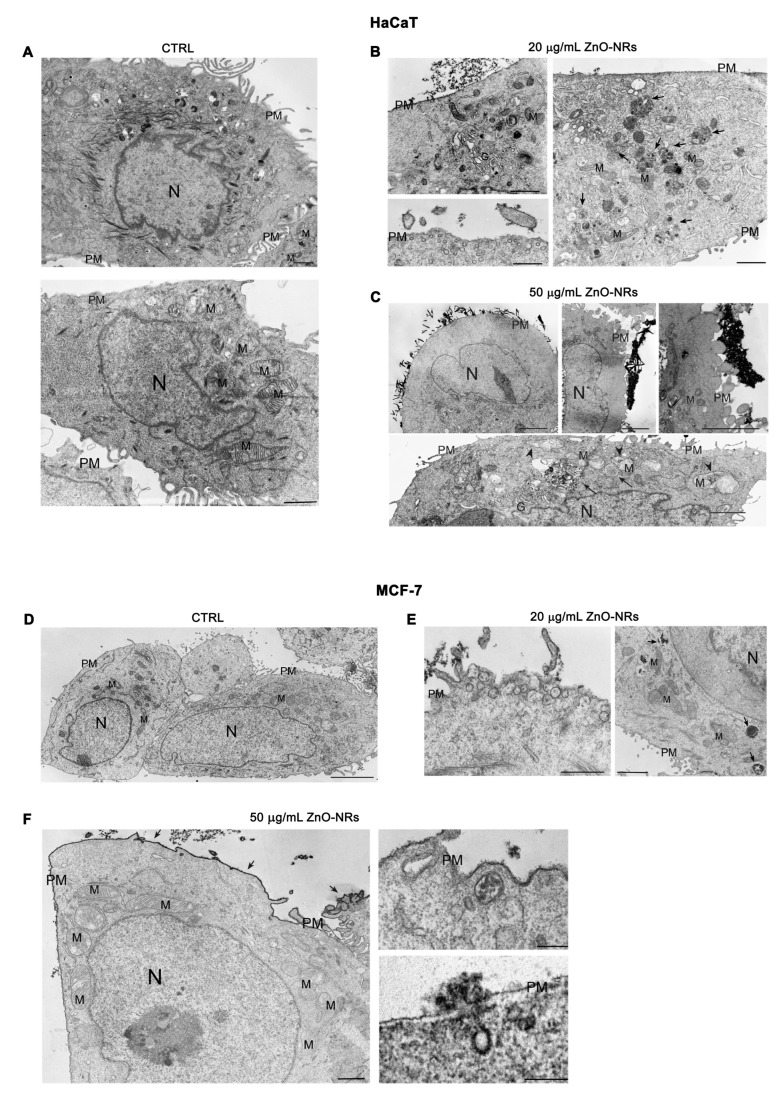
Ultrastructural analysis of HaCaT and MCF-7 cells after ZnO-NR treatment. HaCaT and MCF-7 cells were either untreated or treated with the indicated ZnO-NR concentration for 24 h. (**A**) TEM images of untreated HaCaT cells showing normal cellular morphology. Size bars, 1 µm. (**B**) TEM images of HaCaT cells treated with 20 µg/mL ZnO-NRs showing the extracellular presence and plasma membrane deposition of ZnO-NRs. Arrows point to autophagic vacuoles. Size bars, 1 µm for the upper left and right image and 2 µm for the lower left image. (**C**) TEM images of HaCaT cells treated with 50 µg/mL ZnO-NRs showing increasing extracellular (**upper left**) and intracellular ZnO-NRs accumulation (**lower right**) as well as presence of autophagic bodies (arrows) and damaged mitochondria with disruption/loss of the cristae (arrowhead) (**lower** image). Size bars, 2 µm for the upper image and 1 µm for the lower left and right images. (**D**) TEM image of untreated MCF-7 cells showing cancer cell morphology. Size bar, 5 µm. (**E**) TEM images of MCF-7 cells treated with 20 µg/mL ZnO-NRs showing presence of ZnO-NRs along the microvilli of the plasma membrane, in pinocytic vacuoles localized beneath the plasma membrane (**left** image), and also intracellularly (arrows in **right** image). Size bars, 0.5 µm (**left**) and 1 µm (**right**). (**F**) TEM images of MCF-7 cells treated with 50 µg/mL ZnO-NRs showing deposition along the entire surface of the plasma membrane (arrows) and in intracellular vacuoles. Size bars, 1 µm (**left** image) and 0.2 µm (**right** image). In all panels: N, nucleus; M, mitochondrion; PM, plasma membrane; G, Golgi.

**Figure 4 ijms-24-06971-f004:**
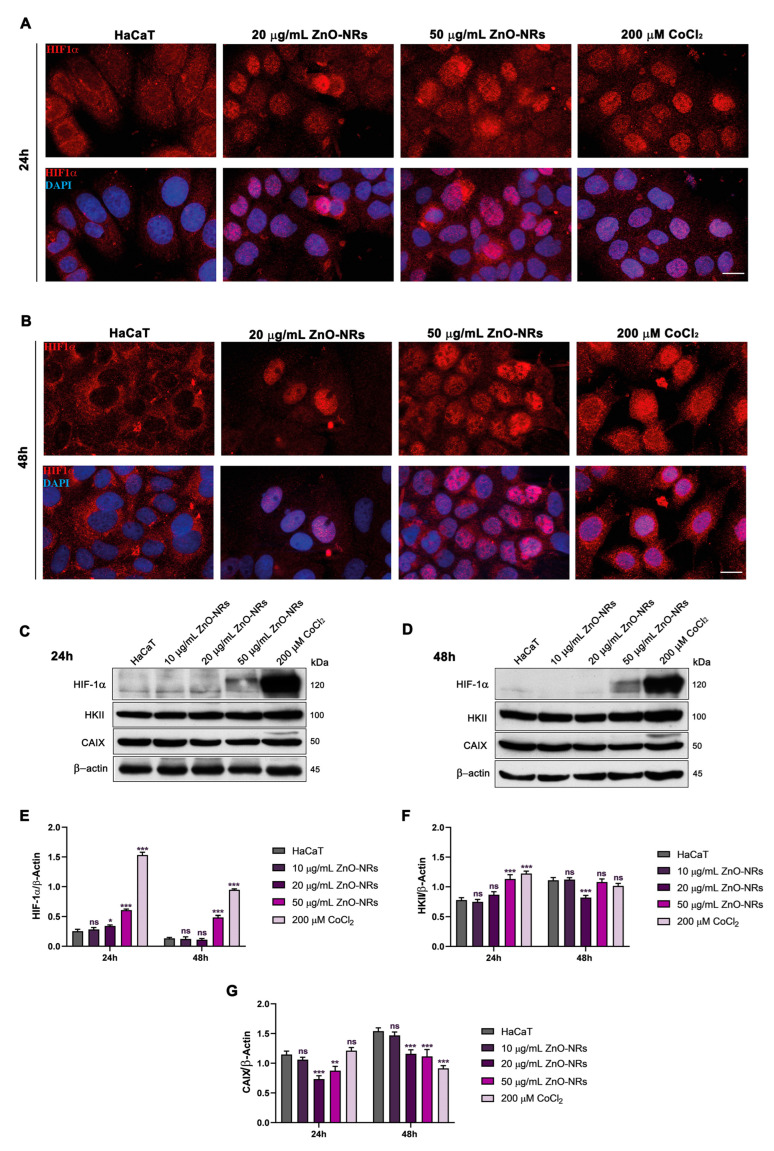
ZnO-NRs’ effects on HIF-1α expression and localization in HaCaT cells. HaCaT cells were either untreated or treated with the indicated ZnO-NR concentration for 24 and 48 h. (**A**) HIF-1α nuclear translocation (red fluorescence) after 24 h treatment with 20 and 50 µg/mL ZnO-NRs. Nuclei were stained with DAPI (blue fluorescence), 40× magnification. (**B**) HIF-1α nuclear translocation (red fluorescence) after 48 h treatment with 20 and 50 µg/mL ZnO-NRs. Nuclei were stained with DAPI (blue fluorescence), 40× magnification. (**C**,**D**) HIF-1α, hexokinase II (HKII), and carbonic anhydrase IX (CAIX) protein expression after 24 and 48 h treatment with 10, 20, and 50 µg/mL of ZnO-NRs. (**E**–**G**) Densitometric analysis of HIF-1α, HKII, and CAIX protein expression after 24 and 48 h treatment with 10, 20, and 50 µg/mL ZnO-NRs. Experiments were repeated three times. * *p* < 0.05, ** *p* < 0.01, and *** *p* < 0.001. ns, not significant. In all experiments, hypoxia, induced by 200 µM CoCl_2_, was used as positive control. β-actin was used as loading control.

**Figure 5 ijms-24-06971-f005:**
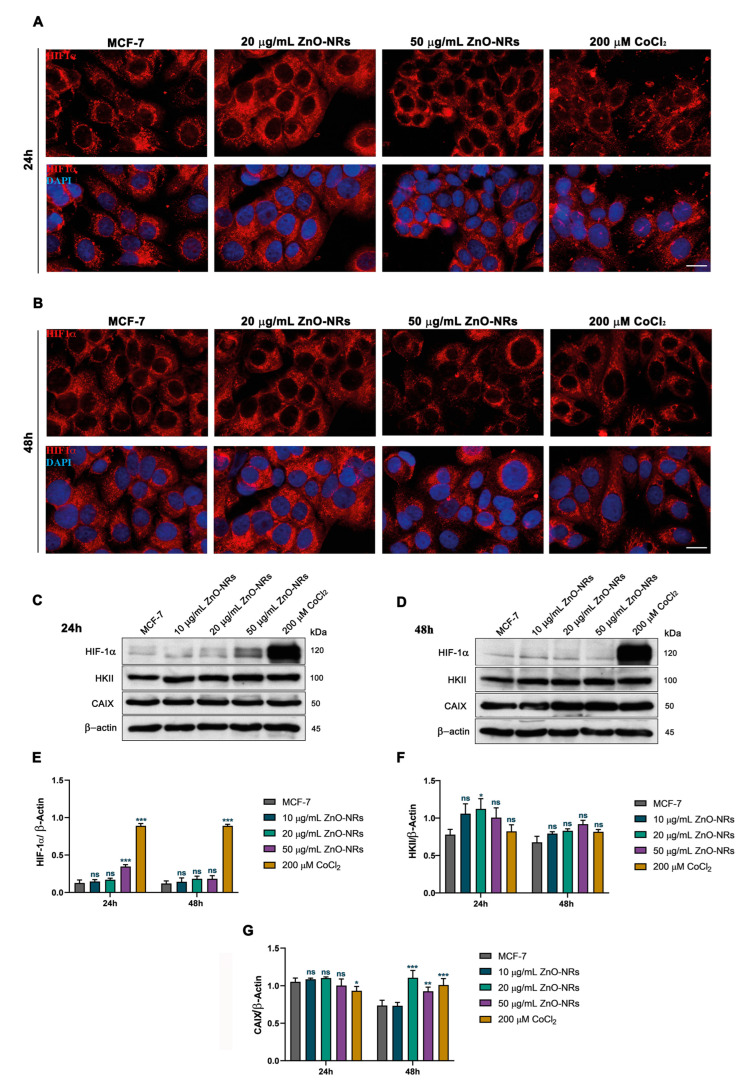
ZnO-NRs’ effects on HIF-1α expression and localization in MCF-7 cells. MCF-7 cells were either untreated or treated with the indicated ZnO-NR concentration for 24 and 48 h. (**A**) HIF-1α nuclear translocation (red fluorescence) after 24 h treatment with 20 and 50 µg/mL ZnO-NRs. Nuclei were stained with DAPI (blue fluorescence), 40× magnification. (**B**) HIF-1α nuclear translocation (red fluorescence) after 48 h treatment with 20 and 50 µg/mL ZnO-NRs. Nuclei were stained with DAPI (blue fluorescence), 40× magnification. (**C,D**) HIF-1α, HKII, and CAIX protein expression after 24 and 48 h treatment with 10, 20, and 50 µg/mL ZnO-NRs. (**E**–**G**) Densitometric analysis of HIF-1α, HKII, and CAIX protein expression after 24 and 48 h treatment with 10, 20, and 50 µg/mL ZnO-NRs. Experiments were repeated three times. * *p* < 0.05, ** *p* < 0.01, and *** *p* < 0.001. ns, not significant. In all experiments, hypoxia, induced by 200 µM CoCl_2_, was used as positive control. β-actin was used as loading control.

**Figure 6 ijms-24-06971-f006:**
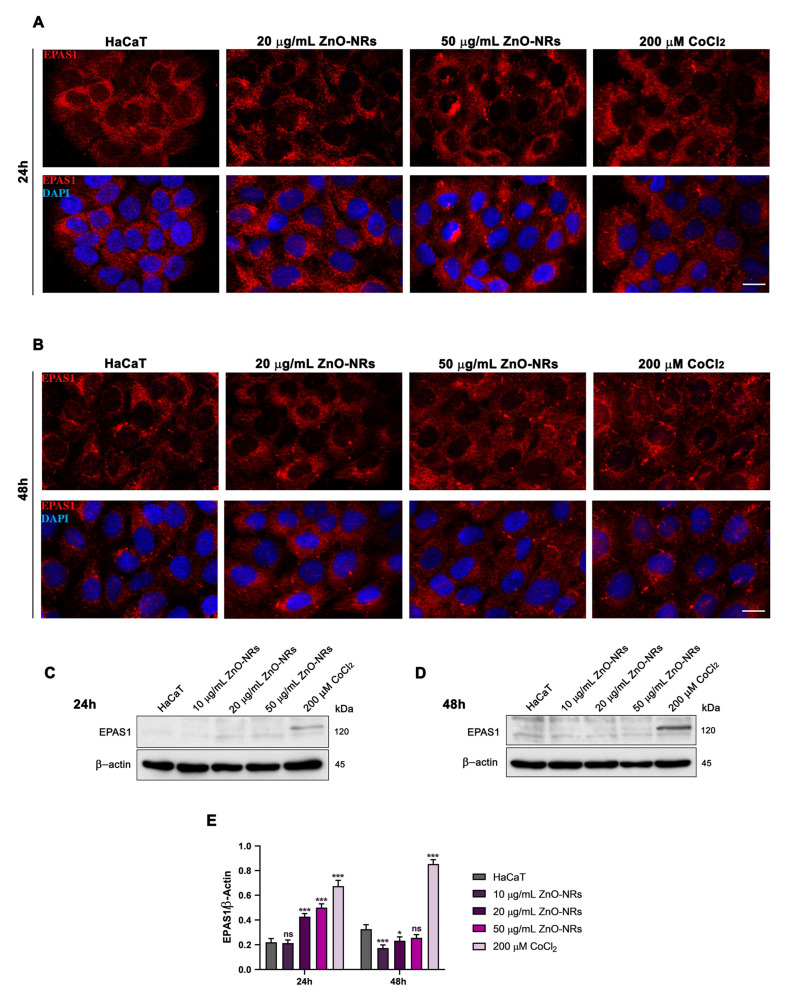
ZnO-NRs’ effects on EPAS1 expression and localization in HaCaT cells. HaCaT cells were either untreated or treated with the indicated ZnO-NR concentration for 24 and 48 h. (**A**) EPAS1 nuclear translocation (red fluorescence) after 24 h treatment with 20 and 50 µg/mL ZnO-NRs. Nuclei were stained with DAPI (blue fluorescence), 40× magnification. (**B**) EPAS1 nuclear translocation (red fluorescence) after 48 h treatment with 20 and 50 µg/mL ZnO-NRs. Nuclei were stained with DAPI (blue fluorescence), 40× magnification. (**C**,**D**) EPAS1 protein expression after 24 and 48 h treatment with 10, 20, and 50 µg/mL ZnO-NRs. (**E**) Densitometric analysis of EPAS1 protein expression after 24 and 48 h treatment with 10, 20, and 50 µg/mL ZnO-NRs. Experiments were repeated three times. * *p* < 0.05, and *** *p* < 0.001. ns, not significant. In all experiments, hypoxia, induced by 200 µM CoCl_2_, was used as positive control. β-actin was used as loading control.

**Figure 7 ijms-24-06971-f007:**
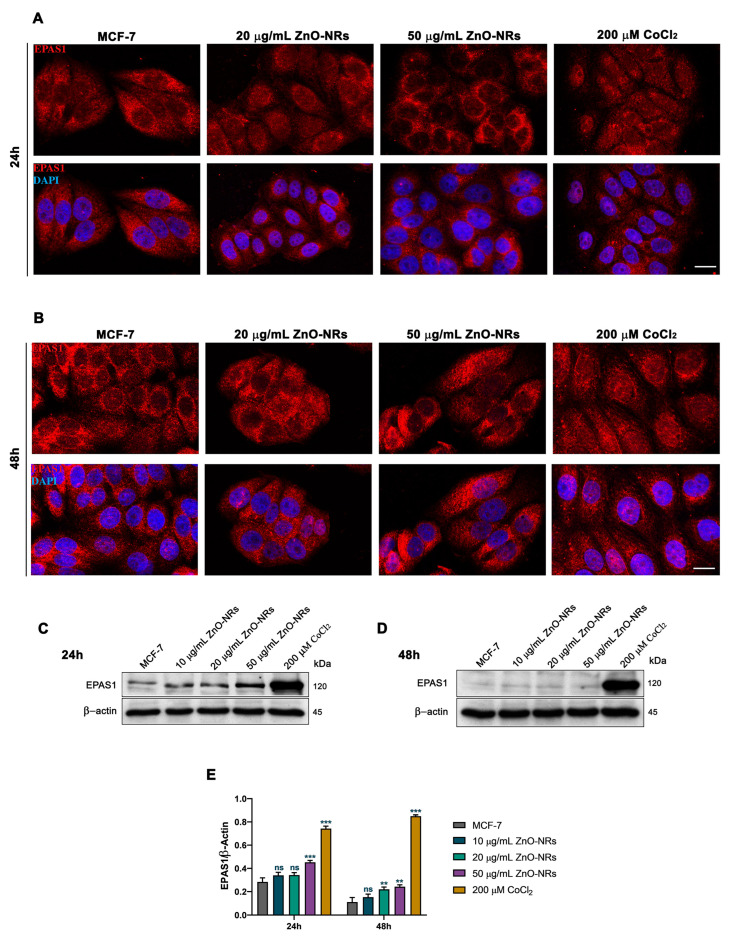
ZnO-NRs’ effects on EPAS1 expression and localization in MCF-7 cells. MCF-7 cells were either untreated or treated with the indicated ZnO-NR concentration for 24 and 48 h. (**A**) EPAS1 nuclear translocation (red fluorescence) after 24 h treatment with 20 and 50 µg/mL ZnO-NRs. Nuclei were stained with DAPI (blue fluorescence), 40× magnification. (**B**) EPAS1 nuclear translocation (red fluorescence) after 48 h treatment with 20 and 50 µg/mL ZnO-NRs. Nuclei were stained with DAPI (blue fluorescence), 40× magnification. (**C**,**D**) EPAS1 protein expression after 24 and 48 h treatment with 10, 20, and 50 µg/mL ZnO-NRs. (**E**) Densitometric analysis of EPAS1 protein expression after 24 and 48 h treatment with 10, 20, and 50 µg/mL ZnO-NRs. Experiments were repeated three times. ** *p* < 0.01, and *** *p* < 0.001. ns, not significant. In all experiments, hypoxia, induced by 200 µM CoCl_2_, was used as positive control. β-actin was used as loading control.

**Figure 8 ijms-24-06971-f008:**
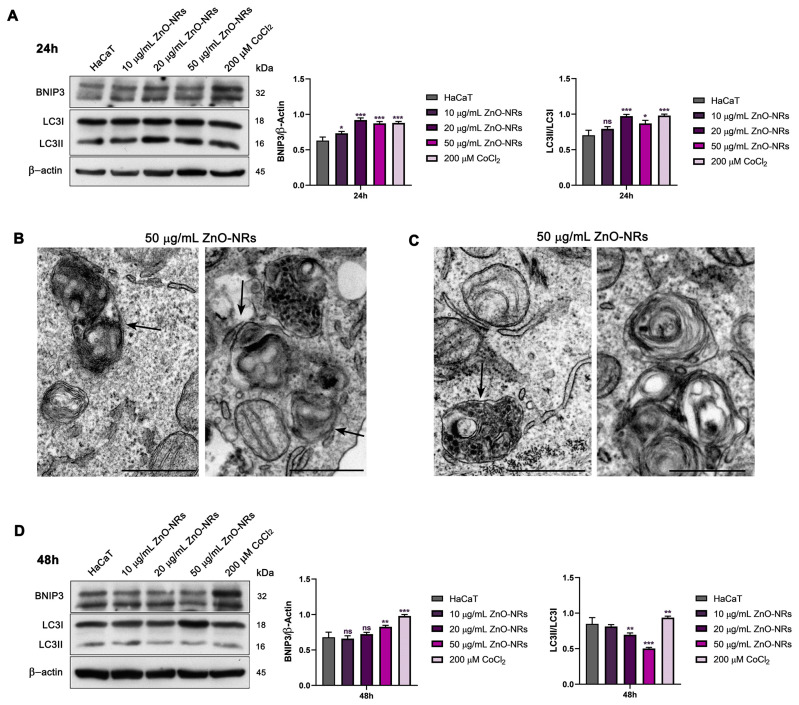
ZnO-NRs’ effects on autophagy and mitophagy in HaCaT cells. HaCaT cells were either untreated or treated with the indicated ZnO-NR concentration for 24 and 48 h. (**A**) BNIP3, LC3I, and LC3II expression after 24 h treatment with 10, 20, and 50 µg/mL ZnO-NRs. Right side, densitometric analysis of BNIP3, LC3I, and LC3II protein expression. (**B**) TEM images of HaCaT cells treated with 50 µg/mL ZnO-NRs for 24 h showing the presence of mitophagic vacuoles (black arrows). Size bars, 0.5 µm. (**C**) TEM images of HaCaT cells treated with 50 µg/mL ZnO-NRs for 24 h showing the presence of autophagic and mitophagic vacuoles (black arrows). Size bars, 0.5 µm. (**D**) BNIP3, LC3I, and LC3II expression after 48 h treatment with 10, 20, and 50 µg/mL ZnO-NRs. Right side, densitometric analysis of BNIP3, LC3I, and LC3II protein expression. (**E**,**F**) Poly(ADP-ribose) polymerase 1 (PARP1) and cleaved PARP expression after 24 and 48 h treatment with 10, 20, and 50 µg/mL ZnO-NRs. Densitometric analysis is reported below each blot. Experiments were repeated three times. * *p* < 0.05, ** *p* < 0.01, and *** *p* < 0.001. ns, not significant. In all experiments, hypoxia, induced by 200 µM CoCl_2_, was used as positive control. β-actin was used as loading control.

**Figure 9 ijms-24-06971-f009:**
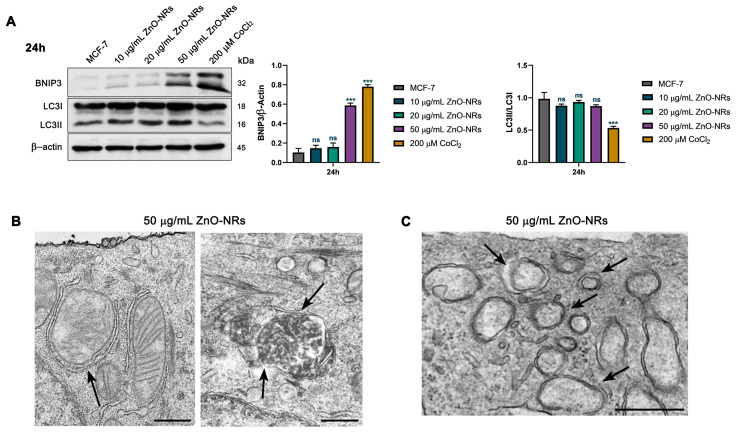
ZnO-NRs’ effects on autophagy and mitophagy in MCF-7 cells. MCF-7 cells were either untreated or treated with the indicated ZnO-NR concentration for 24 and 48 h. (**A**) BNIP3, LC3I, and LC3II expression after 24 h treatment with 10, 20, and 50 µg/mL ZnO-NRs. Right side, densitometric analysis of BNIP3 and LC3II protein expression. (**B**) TEM images of MCF-7 cells treated with 50 µg/mL ZnO-NRs for 24 h showing the presence of mitophagic vacuoles (black arrows). Size bars, 0.5 µm. (**C**) TEM image of MCF-7 cells treated with 50 µg/mL ZnO-NRs for 24 h showing the presence of autophagic vacuoles (black arrows). Size bars, 0.5 µm. (**D**) BNIP3, LC3I, and LC3II expression after 48 h treatment with 10, 20, and 50 µg/mL ZnO-NRs. Right side, densitometric analysis of BNIP3 and LC3II protein expression. (**E**,**F**) PARP1 and cleaved PARP expression after 24 and 48 h treatment with 10, 20, and 50 µg/mL ZnO-NRs. Densitometric analysis is reported below each blot. Experiments were repeated three times. * *p* < 0.05, ** *p* < 0.01, and *** *p* < 0.001. ns, not significant. In all experiments, hypoxia, induced by 200 µM CoCl_2_, was used as positive control. β-actin was used as loading control.

## Data Availability

All data are contained within the article.

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
