# Peer review of "ZnO Nanorods Create a Hypoxic State with Induction of HIF-1 and EPAS1, Autophagy, and Mitophagy in Cancer and Non-Cancer Cells"

_ijms, 2023, doi:10.3390/ijms24086971_

Round 1

Reviewer 1 Report

The article deserves publication in this journal. I have only 3 wishes to the authors.

1) ml should be written mL

2) Figure 9 needs a more detailed legend

3) Separate conclusion required

Author Response

We would like to thank the reviewer for the kind words.

We have changed the manuscript following reviewer’s suggestions as described below.

The article deserves publication in this journal. I have only 3 wishes to the authors.

1)ml should be written mL

We have changed ml into mL throughout the manuscript

2) Figure 9 needs a more detailed legend

We apologize for the lack of information, the legend has been modified into: “FE-SEM images of hydrothermally grown ZnO-NRs as obtained, showing typical dimensions ranging between 40–50 nm in diameter, and lengths of ∼800 nm after sonication.”

3) Separate conclusion required

We have added a Conclusion in a separate section (section 5).

Reviewer 2 Report

Vendor information must be given consistently and completely: e.g. Sigma (St. Loius, MO, USA); Merck (Darmstadt, Germany). Once the location of a supplier has been mentioned, it need not be repeated. Mentioning in the section Materials and Methods Sigma-Aldrich after almost every Chemical you use is annoying. You can avoid this situation with one sentence like: All chemicals used in this study except otherwise stated were purchased…. Milliliters should be written with capital L in the whole manuscript including Figures and graphs. In all figures also use only one font. 

The general rules for writing SI units are following:

The value of a quantity is written as a number followed by a space (representing a multiplication sign) and a unit symbol. This rule explicitly includes the percent sign (%) and the symbol for degrees Celsius (°C). Please change it in the whole manuscript.

Line 320 The sign gr is not a sign for grams but for grains. If you really mean grains please recalculate them for grams. 

Lines 341,  Use decimal points, not commas. Milliliters should be written with capital L.

Line 340 What unit is mMol?

The United States of America is abbreviated U.S.A. or USA.

Missouri can be abbreviated MO.

358 - 359 I would reccomend you to use hard space to avoid this situation.

382, 389 Promega - the vendor information are missing.

413 Please mention version of sotware and reference in possible. I dont understand meaning "National Institutes of Health" in bracket. Please add lacation and expanation.

417 – 422, Please unify name of companies with their locations

How were calculated or measured length and diameter of ZnO-NR. Moreover why images of ZnO-NRs are in the section Materials and Methods? Line 327 and 328 belongs to the Results part.

Please add section Conclusion where will summarized result, impact, novelty, or future perspective of this research and which conclude the research.

Author Response

We would like to thank the reviewer for the suggestions. We have modified the manuscript accordingly as detailed below.

Q:

Vendor information must be given consistently and completely: e.g. Sigma (St. Loius, MO, USA); Merck (Darmstadt, Germany). Once the location of a supplier has been mentioned, it need not be repeated. Mentioning in the section Materials and Methods Sigma-Aldrich after almost every Chemical you use is annoying. You can avoid this situation with one sentence like: All chemicals used in this study except otherwise stated were purchased…. Milliliters should be written with capital L in the whole manuscript including Figures and graphs. In all figures also use only one font. 

A: We have uniformed vendor information. Location of a supplier has been mentioned only the first time. We have added the sentence: “All chemicals and reagents used in this study, except otherwise stated, were purchased from Sigma-Aldrich – MERCK St. Louis, MO, USA” tin the methods to avoid repeating it. Milliliters have been written as mL. We have prepared the Figures using only one font.

Q:

The general rules for writing SI units are following:

The value of a quantity is written as a number followed by a space (representing a multiplication sign) and a unit symbol. This rule explicitly includes the percent sign (%) and the symbol for degrees Celsius (°C). Please change it in the whole manuscript.

A:

We have introduced a space to separate the number and the symbol as suggested in the whole manuscript.

Q

Line 320 The sign gr is not a sign for grams but for grains. If you really mean grains please recalculate them for grams. 

A

We have changed gr into g for grams.

Q

Lines 341,  Use decimal points, not commas. Milliliters should be written with capital L.

A

We changed commas into points and ml into mL

Q

Line 340 What unit is mMol?

A

We apologize for the mistake. We have changed mMol into mM

Q

The United States of America is abbreviated U.S.A. or USA.

A

We have changed US into USA

Q

Missouri can be abbreviated MO.

A

We have abbreviated Missouri with MO

Q

358 - 359 I would reccomend you to use hard space to avoid this situation.

A

We have tried to avoid the situation of lines 358-359 by remodeling the text or adding an hard space.

Q

382, 389 Promega - the vendor information are missing.

A

We have added information for Promega

Q

413 Please mention version of sotware and reference in possible. I dont understand meaning "National Institutes of Health" in bracket. Please add lacation and expanation.

A

We have added the information and the version of the software Image J. This software has been developed at the NIH and is free to download.

Q

417 – 422, Please unify name of companies with their locations

A

We have unified as described above.

Q

How were calculated or measured length and diameter of ZnO-NR. Moreover why images of ZnO-NRs are in the section Materials and Methods? Line 327 and 328 belongs to the Results part.

A

We apologize for the missing information, the measurements were performed by FE-SEM analysis and the text has been modified. “From FE-SEM analysis it resulted that the ZnO-NR have diameters….”. We are sorry for the incorrect placement of Lines 327 and 328 in the methods section and have made the appropriate adjustments in the revised manuscript. We have moved Figure 9, now Figure 1, at the beginning of the Results section.

Q

Please add section Conclusion where will summarized result, impact, novelty, or future perspective of this research and which conclude the research.

A

We have added a Conclusion in a separate section (section 5).